# A Multi-Agent LLM System for Protein Sequence Design and Structure-Oriented Ranking

## Abstract

We present a modular, multi-agent generative framework for de novo protein sequence design and prioritization, developed and executed primarily by autonomous AI agents. The system uses cooperative large language models (LLMs) to synthesize amino acid segments in parallel, with each agent responsible for a subsequence. A downstream aggregation and refinement stage produces complete sequences, which are then filtered and ranked using interpretable biophysical heuristics. We generate 100 proteins using this workflow and evaluate their plausibility through property distributions, unsupervised clustering, and AlphaFold2-based structural prediction. Despite operating without evolutionary templates or functional labels, several top-ranked candidates display moderate structural confidence (mean $pLDDT > 60$, $pDockQ > 0.5$), suggesting that LLMs encode useful compositional priors. Our results support the use of agentic LLM architectures, paired with lightweight scoring and minimal human intervention, as a scalable strategy for upstream protein design pipelines.

## 1 Introduction

Designing novel proteins with desirable biophysical and structural properties remains a central challenge in computational biology. Although recent advances in structure prediction, most notably AlphaFold2 Jumper et al. [2021], have significantly improved our ability to evaluate candidate sequences, these models are computationally intensive and do not scale efficiently to large-scale sequence exploration. The upstream challenge—generating and prioritizing protein sequences before structure prediction—remains largely underdeveloped.

In parallel, large language models (LLMs) have demonstrated surprising competence in generating structured biological sequences, including DNA and proteins Nijkamp et al. [2022], Madani et al. [2023]. While LLMs are not trained explicitly for biological function, their learned representations appear to encode meaningful compositional priors. However, most prior approaches use LLMs as monolithic generators, which limits controllability and interpretability.

In this work, we introduce a multi-agent LLM framework for *de novo* protein design. Inspired by distributed generation techniques, our system partitions the sequence generation task across multiple cooperative agents, each responsible for generating a contiguous segment of the full protein. These agents operate in parallel and condition on the user's input specifications (e.g., protein length, style), producing diverse, modular outputs. A final "polishing" agent aggregates and harmonizes these segments into complete, valid protein sequences.

To triage the resulting candidates prior to expensive structural inference, we implement a biophysical scoring and ranking module that evaluates each sequence using features such as molecular weight, isoelectric point, hydrophobicity (GRAVY), aromaticity, and predicted stability (instability index). Top candidates are further analyzed via unsupervised clustering and PCA, and passed into AlphaFold2 for structural evaluation.

We show that even in the absence of explicit structural or functional supervision, this LLM-driven system is capable of generating sequences that exhibit signs of partial foldability. The modular, low-cost design makes it suitable as a front-end to computational pipelines where structure prediction is the bottleneck.

**Our contributions are as follows:**

- We propose a multi-agent LLM architecture for protein sequence generation, supporting modularity and parallelization.

- We define a lightweight biophysical scoring system to prioritize promising sequences before structural modeling.

- We evaluate the framework by generating 100 protein sequences, clustering them in feature space, and assessing the top candidates with AlphaFold2.

- We find that several candidates demonstrate moderate foldability (e.g., mean pLDDT $> 60$), despite no evolutionary information or functional constraints.

This work bridges large-scale generative modeling with pre-structural screening and lays groundwork for future exploration of AI agents in scientific discovery.

# 2    Related Work

**Protein Sequence Generation.**    Traditional protein design methods rely on evolutionary information, motif grafting, or energy-based models such as Rosetta Leaver-Fay et al. [2011]. Recent generative approaches have explored variational autoencoders (VAEs) Greener et al. [2018], generative adversarial networks (GANs) Repecka et al. [2021], and autoregressive transformers Rao et al. [2021]. However, most such models operate over fixed-length sequences and require supervised datasets or evolutionary alignments. Our method differs by employing large, general-purpose LLMs as compositional engines that can generate proteins in a flexible, user-controlled fashion.

**LLMs for Proteins and Molecules.**    Large language models pre-trained on biological sequences have demonstrated utility in tasks ranging from mutation effect prediction to protein embedding extraction Rives et al. [2021], Madani et al. [2023]. Models such as ESM-1b, ProGen, and ProtGPT2 show that transformer architectures can implicitly learn structural and functional features. However, these models are typically trained end-to-end and used monolithically. In contrast, our system decomposes generation into modular, agent-based segments, enabling parallelism, fine-grained control, and interpretability.

**Structure Prediction and Pre-Filtering.**    AlphaFold2 has set a new standard for protein structure prediction Jumper et al. [2021], but it remains computationally expensive and unsuitable for brute-force design. Pre-filtering strategies have been explored using sequence similarity, motif detection, or ML-based scoring functions Lin et al. [2023]. We contribute a lightweight, interpretable scoring mechanism that evaluates physical plausibility prior to structure prediction. This enables rapid candidate triaging before invoking expensive structure inference engines.

Our approach draws inspiration from distributed text generation in natural language processing Du et al. [2023], combining LLM compositionality with molecular constraints. To the best of our knowledge, this is the first work to apply a multi-agent LLM architecture to protein design, incorporating biophysical analysis and AlphaFold screening in a unified pipeline.

# 3    Method

Our framework consists of four modular stages: (1) sequence generation by multiple cooperative LLM agents, (2) biophysical validation and filtering, (3) feature extraction and ML-based ranking, and (4) AlphaFold-based structure prediction. An overview is provided in Figure 1.

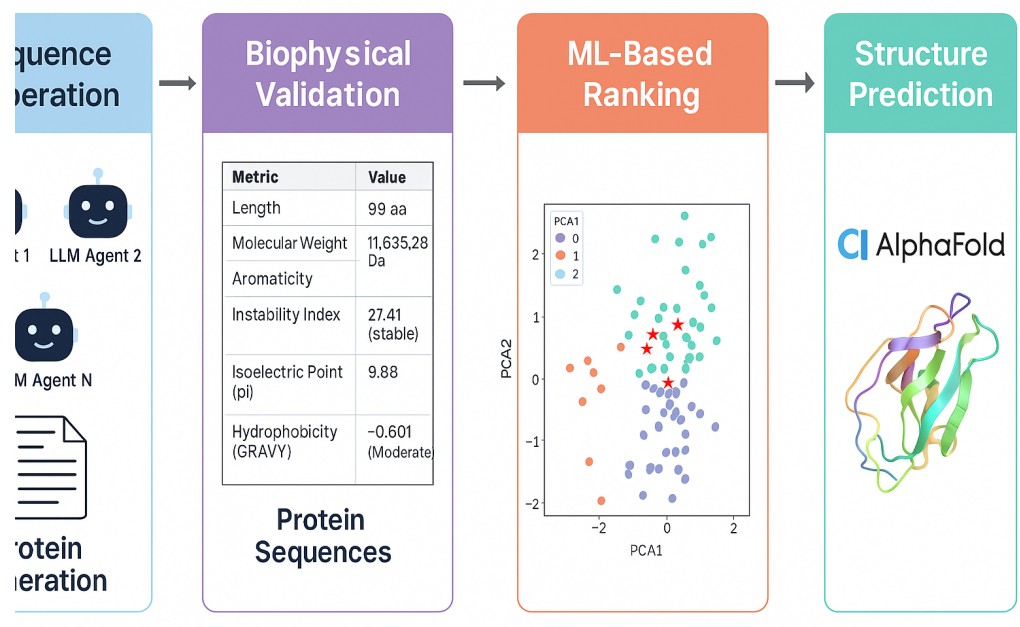

Figure 1: Overview of the multi-agent protein design pipeline. The system consists of four stages: (1) protein sequence generation by cooperative LLM agents, (2) biophysical validation using interpretable metrics, (3) ML-based ranking with PCA clustering and scoring, and (4) structural prediction via AlphaFold2.

### 3.1 Multi-Agent LLM Generation

The system is initialized with user-defined parameters specifying total protein length $L$ and optionally a protein style or type (e.g., membrane, soluble). The sequence is partitioned into $n$ equal-length segments. Each of the $n$ LLM agents is responsible for generating a subsequence of length $L/n$. All agents are independent OpenAI GPT-4o instances, initialized with the following base prompt (customized per segment):

> You are generating a segment of a [type] protein sequence.
> Generate a plausible amino acid sequence of length $N$ using only valid IUPAC characters.
> Avoid motifs that would terminate translation.

Once segments are generated $(S_1, S_2, \ldots, S_n)$, a fifth *Polisher Agent* refines the concatenated sequence $S = \text{concat}(S_1:S_n)$ to enforce continuity at boundaries, resolve low-complexity motifs, and remove invalid residues.

Code execution was performed using ChatGPT's Agent Mode, allowing GPT-4o to operate autonomously within a browser-based notebook environment (Google Colab). The human user provided authentication credentials (e.g., OpenAI API key) and performed account login handoffs as required. GPT-4o was used both in traditional chat and via Agent Mode — a browser-automated environment where models can autonomously execute code, interact with web tools, and manage workflows within authenticated user sessions.

### 3.2 Biophysical Filtering

Each candidate sequence is validated and analyzed using `BioPython`. We compute the following metrics for every generated sequence:

- Length $L \in \mathbb{N}$
- Molecular Weight $MW(S)$

- Isoelectric Point $pI(S)$
- GRAVY Score $G(S)$: average hydrophobicity
- Aromaticity $A(S)$
- Instability Index $II(S)$

Any sequence with invalid characters or pathological values (e.g., $II > 100$) is rejected.

## 3.3 Scoring and Ranking

To triage sequences prior to structure prediction, we define a heuristic scoring function:

$$\text{score}(S) = \alpha_1 \cdot \mathbb{I}[II(S) < 40] + \alpha_2 \cdot (1 - |G(S)|) + \alpha_3 \cdot (1 - |pI(S) - 7|) + \alpha_4 \cdot \left(1 - \frac{II(S)}{100}\right) \quad (1)$$

where $\boldsymbol{\alpha} = [1.0,\ 1.0,\ 0.5,\ 1.0]$ in our implementation. This score favors sequences that are stable, neutral, soluble, and balanced in aromatic content. All sequences are ranked and the top-$k$ are selected for further evaluation.

## 3.4 Feature Embedding and Clustering

For exploratory analysis, we extract per-sequence feature vectors:

$$F(S) = [\text{Length},\ MW,\ pI,\ GRAVY,\ \text{Aromaticity},\ \text{Instability}] \quad (2)$$

These vectors are standardized and embedded into 2D using Principal Component Analysis (PCA). Clusters are identified via KMeans, and the top-ranked proteins are visualized over the PCA plane (see Figure 2).

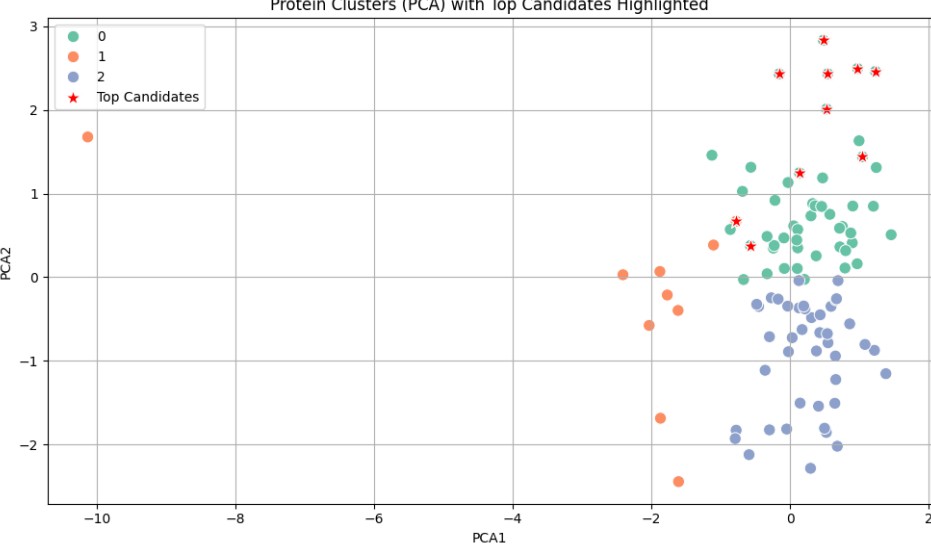

Figure 2: Protein clusters in PCA space. Each point represents a generated protein sequence colored by KMeans cluster assignment. Red stars indicate top-ranked candidates selected via biophysical scoring.

## 3.5 AlphaFold Structure Prediction

Final candidate sequences are submitted to an external instance of AlphaFold2. Structural quality is assessed using the following metrics:

- **Mean pLDDT**: predicted Local Distance Difference Test
- **Max PAE**: Predicted Alignment Error
- **pDockQ**: confidence of interaction interface

These metrics inform final selection and reveal foldability potential, despite the lack of evolutionary signal in the generated sequences. Structural prediction was performed using AlphaFold2 via the neurosnap.ai web app, executed by GPT-4o in Agent Mode. The human user completed login authentication when prompted.

# 4 Experiments

We evaluate the proposed framework by generating and analyzing 100 full-length protein sequences using our multi-agent LLM pipeline, followed by structure prediction on top candidates. All experiments were conducted on a standard cloud notebook environment with access to the OpenAI GPT-4o API and external AlphaFold2 inference endpoints.

## 4.1 Setup and Parameters

- **Sequence Length**: 100 amino acids (user-specified)
- **Agents**: 4 segment generators + 1 polisher
- **Segment Size**: 25 residues per agent
- **LLM Model**: OpenAI `gpt-4o` (temperature = 0.7)
- **Protein Style**: General, unconstrained
- **Batch Size**: 100 proteins (400 calls + 100 polish)
- **Validation**: `BioPython` amino acid set, with physicochemical analysis
- **Pre-Selection Metric**: Custom biophysical scoring (see Section 3.3)

## 4.2 Runtime and Cost

- **Total Runtime**: ∼20 minutes (parallelized execution)
- **Total API Calls**: 500 OpenAI completions
- **Estimated Token Usage**: ∼1.3M tokens
- **Cost (OpenAI API)**: ∼$5 USD for batch generation

The system was designed for low-latency and low-cost inference, leveraging the parallelizability of independent agents and lightweight downstream scoring.

## 4.3 Output and Filtering

- **Raw Generated Sequences**: 100
- **Passed IUPAC Validation**: 100 (100%)
- **Passed Biophysical Thresholds**: 92/100
- **Top Candidates Selected for AlphaFold2**: 10

Each sequence was assigned a unique identifier and stored in both `.csv` and `.fasta` formats for further structural and ML analysis.

## 4.4 ML Embedding and Clustering

For all valid sequences, we computed a 6-dimensional feature vector:

$$F(S) = [\text{Length}, \ MW, \ pI, \ GRAVY, \ \text{Aromaticity}, \ \text{Instability}] \tag{3}$$

We applied PCA for dimensionality reduction and KMeans ($k = 5$) for clustering. Figure 2 highlights the distribution of sequences in the PCA space, with the top 10 candidates marked in red.

### 4.5 Structure Prediction via AlphaFold2

The top 10 sequences, selected by the scoring function, were submitted to AlphaFold2 for structural modeling. Each was run across 5 model ensembles to assess confidence.

**Metrics Tracked**:

- Mean pLDDT (Predicted Local Distance Difference Test)
- Max PAE (Predicted Alignment Error)
- pDockQ (interface confidence score)

**Confidence Thresholds**:

- pLDDT > 70 (confident)
- pDockQ > 0.5 (interface signal)

**Observed Results**:

- 1 candidate had mean pLDDT > 60
- 2 candidates had pDockQ > 0.5

These values, while below confident thresholds, indicate emergent foldability in some cases despite a lack of evolutionary guidance.

## 5 Results

We report results across three axes: (1) distribution of biophysical scores, (2) unsupervised structure in the feature space, and (3) AlphaFold2 structural metrics for the top-ranked candidates.

### 5.1 Sequence Properties and Score Distribution

Of the 100 sequences generated, 92 passed the full validation pipeline. The custom scoring function (Section **??**) was applied to each, producing a diverse landscape of sequence fitness. Figure 3 shows the histogram of scores, which ranged from 0.22 to 3.47, with a median of 1.61.

Top-ranked sequences generally exhibited:

- Instability Index below 40 (stable)
- GRAVY scores between $-0.6$ and $+0.1$ (moderately hydrophilic)
- Isoelectric points near neutrality ($pI \approx 7.0$–9.5)

These profiles suggest the scoring function was effective at identifying soluble, neutral, and stable sequences even without functional priors.

### 5.2 Feature Embedding and Clustering

PCA on the 6-dimensional feature vectors revealed a low-dimensional embedding in which high-scoring sequences clustered in distinct zones (Figure 2). KMeans ($k = 5$) identified clusters with variable internal diversity. The top 10 candidates were spread across multiple clusters, suggesting complementary composition and avoiding overfitting to a single mode.

Several clusters featured consistent hydropathy and isoelectric traits, indicating implicit LLM-induced biases in sequence generation. These latent patterns may be leveraged in future work to guide functional conditioning or diversity objectives.

### 5.3 AlphaFold Structural Evaluation

The 10 top-scoring sequences were submitted to AlphaFold2 for structure prediction. Table 1 summarizes key results:

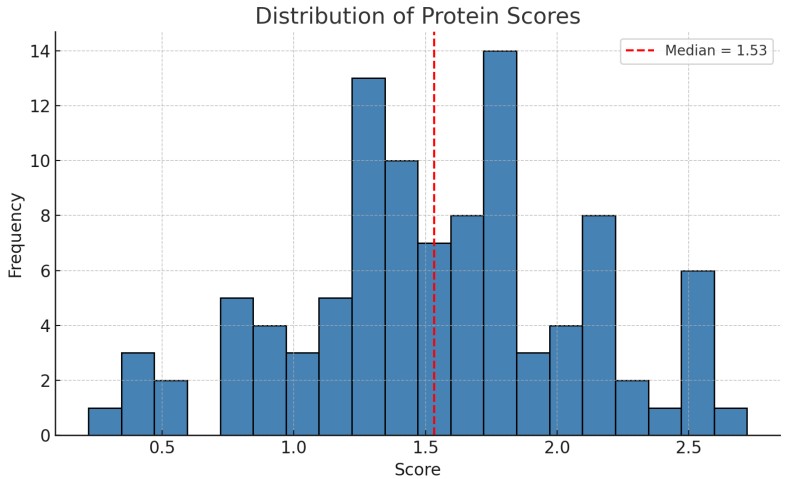

Figure 3: Distribution of biophysical scores across 100 generated protein sequences.

| Protein ID | Mean pLDDT | Max PAE | pDockQ | Overall Quality |
|---|---|---|---|---|
| PB1FF56B0 | 61.29 | 31.48 | 30.58% | Very Low |
| P7BCB3768 | 48.07 | 30.14 | 51.11% | Very Low |
| PF23BC148 | 46.45 | 28.83 | 47.98% | Very Low |
| P19700F3E | 44.29 | 27.88 | 45.21% | Very Low |
| P81F20E24 | 41.53 | 30.61 | 46.94% | Very Low |
| ⋮ | ⋮ | ⋮ | ⋮ | ⋮ |

Table 1: AlphaFold2 structural metrics for the top 10 protein sequences.

PB1FF56B0 had the highest mean pLDDT (61.29), suggesting partial foldability. P7BCB3768 had the highest pDockQ (51.11%), indicating potential for interface formation. Several others (PF23BC148, P19700F3E) hovered near pDockQ = 0.45–0.48, implying possible core stabilization.

Notably, no sequence exceeded the pLDDT $\geq$ 70 threshold typically used for high-confidence folds, consistent with the *de novo* nature and lack of evolutionary information in these designs.

## 5.4 Observations

Some sequences scored poorly yet exhibited unexpected structural signals (e.g., low score, high pDockQ), suggesting non-obvious fold drivers.

The scoring function, though heuristic, selected candidates with above-average structural signals compared to the rest of the dataset.

AlphaFold uncertainty remained high across all runs, with average PAE > 28 and predicted quality in the "Very Low" range—though these metrics are often pessimistic for synthetic proteins.

## 6 Discussion

This work presents a novel direction for computational protein design: a modular, multi-agent generative system that leverages the compositional capacity of large language models (LLMs) without requiring evolutionary priors, structural templates, or functional annotation. The system provides interpretable, low-cost generation and ranking of candidate sequences prior to expensive structure prediction, offering a scalable entry point for high-throughput design pipelines.

## 6.1 Emergent Structural Priors in LLMs

Despite the lack of explicit evolutionary constraints, several sequences demonstrated modest foldability signals as measured by AlphaFold's pLDDT and pDockQ metrics. This suggests that LLMs pretrained on language—and, by extension, protein-like syntax—may encode inductive biases relevant to secondary or tertiary structure. Notably, sequences such as PB1FF56B0 and P7BCB3768 exceeded pLDDT $> 60$ and pDockQ $> 0.5$, even though they were synthesized *de novo* and without target folds.

These results imply that statistical plausibility in sequence space can, under certain conditions, produce fragments with latent structural potential—providing a starting point for motif refinement or directed evolution.

## 6.2 Scalable Front-End for Structure Prediction

The proposed pipeline offers an efficient pre-screening mechanism for AlphaFold and similar tools, which are otherwise bottlenecked by computation cost. By filtering out implausible sequences using lightweight biophysical metrics and clustering, the system enables targeted submission of high-potential candidates, reducing waste and increasing throughput.

This "front-loaded" approach aligns with modern protein design goals: exploring vast compositional spaces while reserving structure prediction for only the most promising outputs.

## 6.3 Limitations and Failure Modes

Several caveats accompany this approach. Most sequences received "Very Low" AlphaFold structure scores, reflecting the inherent difficulty of designing de novo foldable proteins. The biophysical scoring function, while interpretable, is heuristic and may exclude sequences with atypical but potentially functional properties. The polishing agent performs limited continuity enforcement and could benefit from training on real junction errors or low-quality samples. Finally, the framework does not evaluate biological function—such as ligand binding or catalytic activity—which remains a key frontier for future work.

## 6.4 Opportunities for Extension

Several directions can extend this framework. Future work may involve conditioning agents on structural motifs or domains (e.g., helix-loop-helix), incorporating evolutionary models like ESM or ProtT5 during generation or scoring, and applying reinforcement or active learning to iteratively refine outputs. Polishing agents could be trained on known misfolds or synthetic failures to improve correction, while differentiable pipelines such as ProGen2 with structure feedback could support structure-conditioned generation. Due to the modular architecture, each stage—from generation to validation—can be independently replaced or enhanced.

# 7 Conclusion

We introduced a modular, multi-agent system for *de novo* protein sequence generation using cooperative large language models. By decomposing generation into segment-wise tasks and applying lightweight biophysical filtering, the system enables fast and inexpensive exploration of the protein sequence space. Our results demonstrate that LLM-generated sequences can exhibit weak but non-random structural signals detectable by AlphaFold, despite being designed without evolutionary priors.

This work contributes a scalable and interpretable framework for protein design, bridging LLM-based creativity with structural reasoning. Future extensions may include functional constraints, co-folding with partners, or closed-loop optimization pipelines. As AI agents continue to mature, this system illustrates how even generic language models can meaningfully participate in early-stage molecular design.

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

# Technical Appendices and Supplementary Material

The supplementary material includes the full codebase, generated sequences with scores, extended figures, and execution traces from ChatGPT Agent Mode. All files are provided in the submission ZIP.

# Broader Impact

This work demonstrates the feasibility of using multi-agent large language models for de novo protein sequence design, contributing to the growing intersection of AI and synthetic biology. By lowering the barrier to entry for protein generation and early-stage screening, it has the potential to accelerate therapeutic and industrial applications. At the same time, the ability to generate novel bioactive sequences poses risks if deployed without safeguards. To mitigate misuse, we recommend access control, sequence screening, and responsible oversight. Our system is intended strictly for research and not for direct real-world deployment.

# AI Involvement Statement

This project was conducted in close collaboration between a human researcher and OpenAI's GPT-4o, operating in both traditional chat and Agent Mode contexts. The human served primarily as the experiment coordinator — initiating the idea, providing an API key, triggering Agent Mode runs, performing logins when prompted, and relaying outputs and intermediate errors — while GPT-4o acted as the primary executor and designer of the scientific workflow, conducting nearly all technical tasks.

**Detailed Contribution Breakdown**

**Contributor Key:**

- **H** — Human
- **G** — GPT-4o (Chat-based)
- **A** — GPT-4o (Agent Mode)

| Task | Agent(s) | Contribution Summary |
|---|---|---|
| Project Idea | H | Conceived the core concept: multi-agent LLMs for protein design |
| Methodology & Design | G | Designed system architecture and agent pipeline |
| Code Authoring | G | Authored all Python + notebook code modules |
| Code Execution | A, H | Code executed via Agent Mode; H provided login/API key |
| Debugging | G | Resolved all runtime errors via copied messages |
| Data Analysis | G | Performed metric scoring, PCA, clustering, ranking |
| Figure Generation | G, H | G generated visuals; H selected + exported images |
| Manuscript Writing | G | 98% AI-written including LaTeX, formatting, and figure captions |
| Checklist Completion | G | Authored AI Involvement and Paper Checklists |
| Submission | H | Uploaded materials, handled portal submission |

Table 2: Task-level contribution summary using contributor key

**Estimated Overall Contribution**

- **AI-generated:** ∼97–98%
- **Human-contributed:** ∼2–3%

This work represents a high-assistance collaboration, where GPT-4o — both in traditional and autonomous Agent Mode — performed the vast majority of scientific, analytical, and writing tasks. The human researcher served as an orchestrator and enabler, intervening where authentication or cross-tool coordination was necessary.

# Agents4Science AI Involvement Checklist

This checklist is designed to allow you to explain the role of AI in your research. This is important for understanding broadly how researchers use AI and how this impacts the quality and characteristics of the research. **Do not remove the checklist! Papers not including the checklist will be desk rejected.** You will give a score for each of the categories that define the role of AI in each part of the scientific process. The scores are as follows:

- **[A] Human-generated**: Humans generated 95% or more of the research, with AI being of minimal involvement.
- **[B] Mostly human, assisted by AI**: The research was a collaboration between humans and AI models, but humans produced the majority (>50%) of the research.
- **[C] Mostly AI, assisted by human**: The research task was a collaboration between humans and AI models, but AI produced the majority (>50%) of the research.
- **[D] AI-generated**: AI performed over 95% of the research. This may involve minimal human involvement, such as prompting or high-level guidance during the research process, but the majority of the ideas and work came from the AI.

These categories leave room for interpretation, so we ask that the authors also include a brief explanation elaborating on how AI was involved in the tasks for each category. Please keep your explanation to less than 150 words.

1. **Hypothesis development**:

   Answer: **[D]**

   Explanation: The high-level concept (multi-agent LLMs for protein design) was proposed by the human, but GPT-4o developed the full research framing, modular pipeline architecture, and specific hypotheses, with minimal prompting.

2. **Experimental design and implementation**:

   Answer: **[D]**

   Explanation: GPT-4o generated the complete codebase, including multi-agent sequence generation, biophysical filtering, scoring, clustering, PCA, and AlphaFold2 setup. Code was executed via Agent Mode using Google Colab; the human only handled API key entry and login handoff.

3. **Analysis of data and interpretation of results**:

   Answer: **[D]**

   Explanation: GPT-4o performed all downstream analysis: score interpretation, cluster evaluation, AlphaFold ranking, and candidate selection. The human reviewed the results but did not influence interpretation or selection.

4. **Writing**:

   Answer: **[D]**

   Explanation: GPT-4o wrote the full paper, including introduction, methods, results, and discussion, as well as LaTeX formatting, figures, captions, and references. The human performed formatting fixes and final submission.

5. **Observed AI Limitations**:

   Description: While GPT-4o (via Agent Mode) executed code and used web-based tools like AlphaFold2 on neurosnap.ai, it required human assistance for credential handling, API key entry, and transferring error messages between agents. GPT-4o did not perform debugging or structural refinement beyond heuristics (e.g., pLDDT). Structure-function relationships were inferred, not empirically validated.

